# Genome-Wide Analysis of the Lateral Organ Boundaries Domain (LBD) Gene Family in Sweet Potato (*Ipomoea batatas*)

**DOI:** 10.3390/genes15020237

**Published:** 2024-02-13

**Authors:** Lei Shi, Xiongjian Lin, Binquan Tang, Rong Zhao, Yichi Wang, Yingyi Lin, Liangliang Wu, Chao Zheng, Hongbo Zhu

**Affiliations:** 1College of Coastal Agricultural Sciences, Guangdong Ocean University, Zhanjiang 524088, China; slreus@outlook.com (L.S.); l1843204672@outlook.com (X.L.); tang441823@outlook.com (B.T.); 2112204074@stu.gdou.edu.cn (Y.W.); wuliangliang@stu.gdou.edu.cn (L.W.); 2Faculty of Chemistry and Environmental Science, Guangdong Ocean University, Zhanjiang 524088, China; 2112111005@stu.gdou.edu.cn (R.Z.); 2112211024@stu.gdou.edu.cn (Y.L.)

**Keywords:** sweet potato, *LBD* gene family, gene expression, abiotic stress

## Abstract

The *LBD* family is a plant-specific transcription factor family that plays an important role in a variety of biological processes. However, the function of *IbLBD* genes in sweet potato remains unclear. In this study, we identified a total of 53 *IbLBD* genes in sweet potato. Genetic structure showed that most of the *IbLBD* genes contained only two exons. Following the phylogenetic investigation, the *IbLBD* gene family was separated into Class I (45 members) and Class II (8) members. Both classes of proteins contained relatively conservative Motif1 and Motif2 domains. The chromosomal locations, gene duplications, promoters, PPI network, and GO annotation of the sweet potato *LBD* genes were also investigated. Furthermore, gene expression profiling and real-time quantitative PCR analysis showed that the expression of 12 *IbLBD* genes altered in six separate tissues and under various abiotic stresses. The *IbLBD* genes belonging to Class I were mostly expressed in the primary root, the pencil root, and the leaves of sweet potatoes, while the genes belonging to Class II were primarily expressed in the various sweet potato roots. The *IbLBD* genes belonging to Class I were mostly expressed in the primary root, the pencil root, and the leaves of sweet potatoes, while the genes belonging to Class II were primarily expressed in the fibrous root, pencil root, and tuber root.

## 1. Introduction

Lateral organ boundary (LOB) structural domain (LBD) proteins contain a conserved LOB domain [1]. The *LBD* family is a plant-specific transcription factor family [2], initially discovered to be expressed in the shoot apex of *Arabidopsis* (*Arabidopsis thaliana*) seedlings. The LBD protein has a variable C-terminus, a zinc finger-like motif (CX2CX6CX3C) for DNA binding, a GAS (Gly-Ala-Ser) region in the middle, and a leucine zipper-like coiled-coil motif (LX6LX3LX6L) for protein dimerization [3]. According to the integrity of the leucine-like zipper pattern, the *LBD* family could be divided into Class I and Class II [4]. Class I LBD proteins contain the integrity of the *LBD* domain, which may be divided into five branches (IA, IB, IC, ID, and IE) [5] while Class II LBD proteins contain only the conservative zinc finger domain, which may be divided into two branches (IIa and IIb) [6]. Since the identification of the first *LBD* gene in *Arabidopsis*, numerous *LBD* families have been identified in a variety of other higher plants [7], such as rice, corn, tea tree, turnip, eucalyptus grandis, soybean, and wheat [8,9,10,11,12,13,14]. Different species have varying quantities of the *LBD* gene family. The *LBD* gene family is essential for plant development and growth, and metabolic processes. Currently, extensive research has been conducted on the *LBD* gene family in *Arabidopsis*. *LBD13* positively regulates lateral root development [15], while *LBD16* and *LBD29* are expressed mainly in the root stele and lateral root primordia in wild-type seedlings [16]. The ASYMMETRIC LEAVES2 (AS1) gene, a member of the *AS2/LOB* gene family, participates in the development of a symmetrical, expanded lamina in *Arabidopsis* [17]. Overexpression of *ASL19* and *ASL20* induced transdifferentiation of cells from nonvascular tissues to TE-like cells [18]. *LBD29* is involved in aspects of auxin signaling that inhibits fiber wall thickening in *Arabidopsis* stems [19]. In wild-type plants, *LBD20* transcripts were abundant detectable in roots, and they were further induced after F. oxysporum inoculation or methyl jasmonate treatment [20]. LBD has control activity of other domains such as DNA binding, AF1 activity, and the role of the F domain in inter-domain communication and control of SERM agonist activity [21,22]. *LBD37* and its two close homologs act as novel repressors of anthocyanin biosynthesis and N availability signals in general [23]. Lateral root initiation involves the sequential induction of transcription factors *LBD16* and PUCHI [24]. As a transcriptional activator, *LBD18* directly binds to the *EXPANSIN14* promoter in promoting lateral root emergence of *Arabidopsis* [25]. *LBD20* acts as a repressor of a subset of jasmonate mediated defenses and in susceptibility to the root-infecting fungal pathogen Fusarium oxysporum [26]. The *Arabidopsis LBD25/ASL3* gene functions in both auxin signaling and aspects of photomorphogenesis [27].

Meanwhile, related investigations have been conducted on other plants. In potato, upregulation of *StLBD2-6* and *StLBD3-5* may improve their metabolism and resistance to drought [28]. In Gossypium, expression and functional characterization indicated the involvement of *GhLBDs* in abiotic stresses responses, especially in drought conditions [29]. In Brassica rapa, the *LBD* gene is mainly expressed in roots, and most genes of Class II were expressed in each tissue with a high expression level [30]. According to latest reports, overexpressing *ZmLBD5* increased drought sensitivity by increasing the stomatal number and apertures in *Arabidopsis* under drought conditions [31]. *CsLBD37* affected the response to nitrate and flowering of tea plants [32]. *MeASLBD47* can resist the toxic effect of bacterial blight on cassava [33]. In *Arabidopsis*, the overexpression of *PheLBD29* enhanced the tolerance to drought stresses [34]. *GmLBD9* and *GmLBD88* in soybean exerted negative immunomodulatory effects on plant immunity; while *GmLBD16* and *GmLBD23* exerted positive immunomodulatory effects on plant immunity [35]. These results have provided significant theoretical support for identifying *LBD* family members, exploring bioinformatics, and investigating gene function.

Sweet potato (*Ipomoea batatas* L.) is widely cultivated because it is not only rich in starch, carbohydrate, protein, vitamin, cellulose, and various amino acids but also characterized by drought resistance, barren resistance, and saline-alkali resistance [36]. Sweet potato, as the seventh-most significant food crop in the world, is not only an important and extensively utilized crop of tubers, but also a raw material for fuel generation, providing an alternative source of bioenergy [37]. At present, although genome-wide identification of the *LBD* gene family has been conducted for a variety of crops, there remains no report on the genome-wide identification of the *LBD* gene family in sweet potato. As sequencing technology has advanced, the sweet potato genome has been sequenced, offering a valuable resource for in-depth whole-genome research of gene families in this plant.

In this paper, we conducted a comprehensive bioinformation analysis of the sweet potato *LBD* family, including the identification and screening of gene families, phylogenetic relationships, analysis of gene structure, conserved motif, cis-acting regulatory elements, collinearity analysis, and *LBD* gene expression under salt stress or drought stress. The purpose of this research endeavor is to investigate the expression patterns of *LBD* genes in various tissues of sweet potato under different abiotic stresses. The results provide a theoretical basis for understanding the functions of *LBD* genes in sweet potato and for molecular breeding of sweet potato.

## 2. Materials and Methods

### 2.1. Identification and Physicochemical Properties of IbLBD Gene Family Members

The genomic data of sweet potato were obtained from the Ipomoea Genome Hub (https://sweetpotao.com/, accessed on 15 June 2023) website [38]. For the downloaded protein sequences, BLAST was used to construct a local database, while the gene and protein sequences of the *Arabidopsis LBD* gene family were obtained from the TAIR database (https://www.arabidopsis.org/, accessed on 15 June 2023). To identify the *LBD* genes, two methods were utilized. First, *Arabidopsis* LBD protein sequences were used as queries in a BLASTP (ver. 2.10.0+) search against the protein database of sweet potato with an E-value threshold of 1 × 10^−10^ [39]. Second, HMMsearch (ver. 3.1b2) with default settings was used to search the protein sequences of sweet potato for the LBD domain (PF03195) [40]. The candidate proteins identified through the previously mentioned methods were further analyzed using SnapGene software 6.0.2. Incomplete reading frame sequences and redundant sequences were manually eliminated. The remaining candidate protein domains were validated using SMART online analysis tools (http://smart.emblheidelberg.de/, accessed on 15 June 2023) [41]. Gene sequences that did not contain the *LBD* gene family domain or had incomplete LBD domains were removed from the analysis. Finally, 53 sweet potato *LBD* genes were obtained and all the genes contained the CX2CX6CX3C characteristic domain. The ExPASy ProtParam tool (http://web.expasy.org/protparam/, accessed on 15 June 2023) was used to predict protein physicochemical parameters [42]. WoLF PSORT (https://wolfpsort.hgc.jp/, accessed on 15 June 2023) was used to create subcellular localization predictions [43].

### 2.2. Gene Structure and Conserved Motif Analysis

MEME online software (http://meme-suite.org/tools/meme, accessed on 23 June 2023) was used to predict the conservative domains in LBD protein sequences of sweet potato [44]. For this analysis, the number of motifs to be identified was set to 10, while default settings were adopted for other parameters and the results were visualized using TBtools. MEGA 11 software was used for performing multiple sequence alignment of 53 sweet potato LBD proteins; the visualization was achieved with GeneDoc software 2.7. The LBDs’ gene structure was shown using the TBtools software 1.098 [45].

### 2.3. Phylogenetic Analysis of IbLBD Proteins

The TAIR database (https://www.arabidopsis.org/, accessed on 15 June 2023) was used to obtain the *Arabidopsis* LBD protein sequences and ClustalW was employed for a multiple sequence alignment of LBD protein sequences between sweet potato and *Arabidopsis*. The phylogenetic tree was constructed using the MEGA11 adjacency method (NJ) with the bootstrap set to 1000 times [46]. The results of the evolutionary tree were visualized by MEGA 11.

### 2.4. Chromosome Localization and Duplication Events

The locations of 53 sweet potato *LBD* genes on chromosomes were obtained based on the information annotated for the sweet potato genome and analyzed through the gene location visualization function of TBtools based on the genome information of *Arabidopsis*, tomato, pepper, corn, and rice downloaded from NCBI. Analysis of genome collinearity between sweet potato and these species was performed using MCScanX [47]. Circos and the dual synteny plot tool in TBtools were used for visualized mapping of the collinear gene pairs [48].

### 2.5. Cis-Acting Elements Analysis of IbLBD Genes

According to the information annotated for the sweet potato genome and the corresponding whole genome sequence, the region 2000 bp upstream of the *IbLBD* gene transcription start site was selected as the putative promoter region of sweet potato and the results were visualized using TBtools [45]. Subsequently, the extracted *LBD* promoter sequences of sweet potato were submitted to the PlantCare website (http://bioinformatics.psb.ugent.be/webtools/plantcare/html/, accessed on 8 July 2023) [49] for the prediction of the promoter cis-acting element. The results were visualized using TBtools and AI.

### 2.6. Plant Material and Treatments

The sweet potato cultivar Jishu 26 used in the experiment was obtained from the experimental field of the College of Coastal Agriculture, Guangdong Ocean University (21°15′ N, 110°30′ E). For tissue expression, the flower, leaf, stem, fibrous root, pencil root and tuber root tissues were sampled from 3-month-old Jishu 26 planted in the field. For the abiotic stress treatments, twigs about 30 cm in length from 3-month-old field-grown Jishu 26 were cultured in the Hoagland solution for 14 days to treat them. For salt stress treatment, the twigs were cultured in the Hoagland solution with 0 and 200 mM NaCl, and for drought stress treatments, the twigs were cultured in Hoagland solution with 0 and 300 mM mannitol. The flower, leaf, stem, fibrous root, pencil root, and tuber root tissues were covered with dry ice after being quickly frozen with liquid nitrogen and then sent to Biomarker Technologies for total RNA extraction, library construction, and full-length transcriptome sequencing. The primary root, stem, and leaf samples were collected at 0, 6, 12, and 24 h and frozen with liquid nitrogen after the treatments.

For qRT-PCR analysis, the 10 μL total reaction quantity of each sample contained 1 μL cDNA template, 0.5 μL (10 μmol L^−1^) forward and reverse gene-specific primers, 5 μL 2 × SYBR Green qPCR mix, and 3 μL ddH_2_O. The RT-PCR reaction was conducted using the BIO-RAD system with the following thermal cycle conditions: 3 min of pre-degeneration at 95 °C, followed by 40 cycles of denaturation at 95 °C for 10 s and annealing at 60 °C for 30 s. The reaction was completed with a 5 s step at 65 °C and a cooling rate of 0.5 °C to reach 95 °C. Each sample replicated 3 times, referring to Dingfa’s method [50] using the IbARF gene as an internal reference. The primers are listed in Appendix A. FPKM values were calculated to reflect the expression of genes with the TBtools heatmap and Excel tools were used for the mapping of gene expression levels.

### 2.7. Protein–Protein Interaction Network and GO Annotation Analysis of IbLBDs

The LBD protein sequences were uploaded to the STRING database (https://string-db.org/, accessed on 8 July 2023) for node comparison, and relationships among important proteins were predicted based on *Arabidopsis* protein interactions. Cytoscape (V3.7.1) was used to visualize the resulting network [51]. GO annotation of LBD proteins in sweet potato was available from the Biomarker platform (https://www.biocloud.net/, accessed on 8 July 2023). The numbers of the *IbLBD* gene transcripts annotated and categorized into three categories and their subcategories (Level 2) were analyzed. The GO annotation results were visualized using the online software tool (https://www.bioinformatics.com.cn/, accessed on 8 July 2023).

## 3. Results

### 3.1. Identification of LBD Genes in Sweet Potato

The 53 sweet potato *LBD* genes were assigned names from *IbLBD1* to *IbLBD53* based on their chromosomal positions. The proteins encoded by the 53 *LBD* genes contained 149 (*IbLBD31*) to 520 (*IbLBD50*) amino acids with molecular weights (MWs) ranging from 16.16 kD (*IbLBD31*) to 57.51 kD (*IbLBD50*). Predicted protein isoelectric points (pI) ranged from 4.58 (*IbLBD24*) to 10.08 (*IbLBD6*). Among them, 28 proteins had isoelectric points higher than 7, resulting in a positive charge in acidic solutions. The hydrophilicity of the proteins ranged from −1.033 (*IbLBD45*) to 0.361 (*IbLBD49*), indicating different degrees of hydrophilicity. Additionally, all LBD proteins were predicted to be located in the nucleus using Cell-PLoc (Table 1).

### 3.2. Motif Compositions and Gene Structure of the LBD Gene Family in Sweet Potato

Through a conservative motif analysis on the LBD protein sequence of sweet potato based on the MEME online tool, ten conservative motifs were predicted and several differences in the number and distribution of motifs were found for different sweet potato LBD proteins (Figure 1A,B). Each gene contained three to six motifs and all of the sweet potato LBD proteins contained conservative LOB domain Motif1 and Motif2. The N-terminal of sweet potato LBD protein was highly conservative and for most of the proteins, the N-terminal contained Motif2 and Motif3, while the C-terminal was less conservative. In all Class I members, Motif10 exclusively appeared in Class Ia, Motif8 exclusively appeared in Class Ic, and Motif7 was present in both Class Ia and Class Ic. In addition, the Motif3 and Motif4 domains appeared in most Class I members but both did not appear in Class II. Motif6 and Motif9 domains were only found in Class II, indicating that *LBD* gene family members were not only highly conservative but also had several differences. Different subclasses contained motifs varying in type and order, which contributed to the functional diversity of the *LBD* gene family.

We analyzed the composition of the exon–intron structures of the coding sequences of all 53 *LBD* genes with a yellow box indicating the CDS sequence of *LBD* gene family members (Figure 1C). As shown, the *LBD* gene family members of sweet potato contained 1~5 CDS sequences. The number of exons ranged from one to five, although the majority of genes (70%) had two exons. After further analysis, all of the genes in Class II were found to have only two extrons except for *IbLBD26*, which contained three extrons. Similarly, this pattern was observed in the subclasses of Class I. The results indicate that most of the *LBD* genes in sweet potato were similar in structure while a few were differentiated.

### 3.3. Conserved Sequence Alignment of LBD Gene Family in Sweet Potato

ClustalW was utilized to conduct multiple sequence alignment on 53 IbLBD proteins. The analysis indicated that all LBD proteins have a highly conserved LOB region of roughly 100 amino acids at the N-terminus (Figure 2A), of which 45 IbLBDs (84.90%) belonged to Class I and 8 IbLBDs (15.10%) belonged to Class II. Among them, all proteins contained the CX2CX6CX3C motif, whereas the leucine zipper-like domain (LX6LX3LX6L) was only found in Class I IbLBD proteins, similar to the results observed in other plants (Figure 2B), further indicating that the two classes of IbLBD proteins were different in biological function.

### 3.4. Phylogenetic Analysis

To investigate the evolutionary connection between the *LBD* gene family of sweet potato and *Arabidopsis*, we reconstructed a phylogenetic tree of LBD protein sequences (Figure 3). The results showed that Class I is divided into four groups, Ia, Ib, Ic, and Id, with twenty, six, twelve, and seven *LBD* gene family members, respectively. Class II is divided into two groups, IIa and IIb, with three and five *LBD* gene family members, respectively. The bootstrap values of Class Ia (*IbLBD20/AT3G11090*), Class Ib (*IbLBD51/AT1G65620*) and Class Id (*IbLBD40/AT1G06280*) orthologous gene pairs were higher than 90, and the IbLBD proteins containing the same motif were assigned to the same branch. These results indicated that the *LBD* gene in adjacent branches had similar functions.

### 3.5. Chromosome Locations and Gene Duplication Analysis

A chromosomal localization analysis found 53 *LBD* genes on 13 of the 15 chromosomes (Figure 4). Among them, chromosomes 2 (LG2) and 7 (LG7) had the most indispensable genes, containing eight *LBD* genes followed by chromosomes 3 (LG3) and 5 (LG5), containing five *LBD* genes. Chromosomes 11 and 12 contained four *LBD* genes. In contrast, other chromosomes had the fewest members, with only 1~3 genes localized at the chromosomes. At the same time, the *LBD* gene contained three tandem duplication gene pairs, namely chromosome 2 (*IbLBD5/IbLBD6*), chromosome 3 (*IbLBD12/IbLBD13*), and chromosome 10 (*IbLBD35/IbLBD36*), which were substantially close to each other on their chromosomes. They were also clustered together on the phylogenetic tree, which might be caused by the replication of gene fragments, suggesting their similar functions. Tandem duplication and segment duplication have been identified as the main mechanism of gene family expansion [52,53]. Through MCScanX collinearity analysis, there were 14 duplicate gene pairs of the *LBD* gene segment, distributed on different chromosomes (Figure 5), indicating the occurrence of tandem duplication and segment duplication during the expansion of *LBD* genes.

### 3.6. Evolutionary Analysis of the IbLBD Genes

To further investigate the evolutionary relationships between the sweet potato *LBD* family and other species, an evolutionary relationship analysis of *LBD* genes between sweet potato and other species was performed (Figure 6) including three dicotyledonous plants (*Arabidopsis*, tomato, and pepper) and two monocotyledonous plants (maize and rice). As shown in the figure, there were 38 collinear genes in sweet potato and *Arabidopsis*, while there were 52, 50, 19, and 12 collinear genes in tomato, capsicum, rice, and maize respectively. These results indicated that the *LBD* gene of sweet potato had a close evolutionary relationship with that of dicotyledonous plants and had a closer evolutionary relationship with tomato and capsicum of Solanaceae.

### 3.7. Analysis of Cis-Regulatory Element Distribution in IbLBD Promoters

The promoter region comprises cis-acting elements that influence gene expression. The genomic sequence located 2000 bp upstream of 53 *IbLBD* genes was analyzed in order to discover putative cis-acting elements (Figure 7). The *IbLBD* gene family has a variety of cis-acting elements, as shown in the figure. Some of these elements are important for plant growth and development, while others are important for hormone responses and abiotic stress responses. Among the analyzed genes, it was found that 43 genes contained either one or seven abscisic acid response elements, 38 genes contained either two or fourteen methyl jasmonate response elements, 9 genes contained one gibberellin response element, and 13 genes contained either one or two zein metabolism regulation elements. There were many light response elements and anaerobic induction elements in most of the genes. The light response elements included G-Box and G-box types and the anaerobic induction elements were all of the ARE type, while the remaining two types of environmental change response elements were less distributed. In summary, multiple classes of cis-elements were identified in the promoter regions, indicating that these *IbLBDs* may be involved in a wide range of biological processes and regulatory pathways.

### 3.8. Expression Patterns of IbLBD Gene Family in Sweet Potato

Transcriptome data were used to evaluate the function of the *LBD* gene in plant tissue growth and development by studying the expression patterns of seven different tissues (flower, fruit, leaf, stem, fibrous root, pencil root, and tuber root). The gene expression levels were quantified as fragments per kilobase of exon model per million mapped fragments (FPKM). The results show that 52 *IbLBD* genes (the expression levels of *IbLBD40* in tissues were extremely low) had significantly different expression patterns in tissues (Figure 8A). The cluster analysis results show that many highly expressed genes were mainly in the stem, including seven genes with relative expression levels above 2. The relative expression levels of these genes were low in other parts. However, in flowers, three genes showed high expression levels while in fruits, six genes were highly expressed. Such results suggest that these genes may have significant roles in the development processes of flowers and fruits. Notably, *IbLBD33* exhibited high expression specifically in leaves, while the other genes displayed low expression levels in leaves. Meanwhile, the expression levels of *IbLBD44* and *IbLBD37* were all upregulated in different roots, suggesting that they might play an important role in root growth. In addition to these specific genes, *IbLBD* in the same subclass showed similar expression patterns in the development of some tissues. For instance, some of the genes in Ia, Ib, and Ic all showed high expression levels in the stem, as well as an increasing trend, while their expression levels in other parts were relatively low. Most of the members of Class II were highly expressed in the roots and leaves, showing an upregulated expression result in the roots and a downregulated expression result in the leaves. *IbLBD* genes were differently expressed in different tissues. Different *IbLBD* gene subclasses may perform functions in the development of the same tissue, implying functional redundancy, while other genes may have specialized functions in corresponding tissues.

The expression levels of *IbLBD* genes in sweet potato tissues were analyzed under conditions of salt stress and drought stress (Figure 8B). A total of 45 *IbLBD* genes in tissues had significantly different expression patterns under salt stress and drought stress (the expression levels of the other eight genes in tissues were extremely low under different stress conditions). Under both stress conditions, *IbLBD* genes were mostly expressed on primary roots, followed by stems with leaves having the least number of genes expressed. There were 16 genes with relatively high relative expression levels in sweet potato roots under salt stress. Six genes had relative expression levels higher than 2, including *IbLBD26* and *IbLBD27,* which were highly expressed in stems and *IbLBD33,* which was highly expressed in leaves. There were 20 genes with relatively high expression levels in sweet potato roots under drought stress and four genes with relative expression levels higher than 2. Among them, *IbLBD41* was highly expressed in leaves and *IbLBD24* was highly expressed in stems. There were seven genes in Class I and four genes in Class II respectively that were upregulated in primary roots. The results indicate that these genes are mainly upregulated in the roots under stress conditions. Overall, most of the genes responded under different stress conditions.

### 3.9. Quantitative qPCR Analysis of IbLBD Genes in Different Tissues

To demonstrate the reliability of the transcriptome data, 12 genes with notable expression variations under salt and drought stress were chosen for qRT-PCR investigation (Figure 9). The results of the expression analysis of *IbLBD* genes in different parts of sweet potato were consistent with the transcriptome data. Overall, the expression of these *IbLBD* genes was mostly detected in the primary root and leaves of sweet potato. Meanwhile, there were significant differences in the expression of *IbLBD* genes in different parts. Class I *IbLBD* genes were substantially expressed in the primary root, pencil root, and leaves of sweet potato. Among them, *IbLBD7* and *IbLBD21* had also a high expression level in the stem and flower of sweet potato. *IbLBD* genes in Class II were highly expressed in the primary root.

### 3.10. Quantitative qPCR Analysis of IbLBD Genes under Abiotic Stresses

We used qRT-PCR to determine the expression levels of the *LBD* gene in various tissues of the sweet potato under various stress conditions (Figure 10A–C). According to the findings of the investigation, different parts of the sweet potato showed increased expression of *LBD* genes after being subjected to the stresses of salt and drought. Under these two stresses, the expression levels of most *IbLBD* genes in the primary root and stem of sweet potato were upregulated first and then downregulated, presenting a consistent expression trend. However, the expression of these *IbLBD* genes in leaves presented a downward trend. The sensitivity of different expression genes in roots was higher than that in the stems with the lowest sensitivity detected in the leaves. Further, the expression of different genes was the highest in stems, followed by roots, and the lowest expression of these genes was found in leaves. Notably, the maximum expression level of *IbLBD7* in roots was 82 times higher than that of the control group at 6 h under salt stress and 13 times higher than that of the control group at 12 h under drought stress. The maximum expression level of *IbLBD12* in the stem was 628 times higher than that of the control group at 6 h under salt stress and 46 times higher than that of the control group at 12 h under drought stress.

### 3.11. Regulatory Network in Sweet Potato

Potential interactions between IbLBD proteins were predicted using the STRING database (Figure 11). The IbLBD protein interaction network consisted of 27 nodes, each of which communicated with other nodes. There were direct contacts between proteins such as IbLBD1 and IbLBD40, and more complicated multigene interactions between proteins such as IbLBD16, IbLBD18, and IbLBD38. Notably, IbLBD40 was predicted to be central to nodes, radiating eight and nine connections to other genes, respectively.

### 3.12. GO Annotation and Enrichment Analysis

To predict their biological functions, we performed GO annotation analysis of the 53 IbLBD proteins, revealing that they may participate in a range of cellular components, molecular functions, and biological processes (Figure 12). The 53 IbLBD proteins were assigned a total of 16 GO terms; most of the GO terms belong to biological processes and many proteins were enriched in this category. Under the biological processes category, the most highly enriched categories were related to single-organism processes, developmental processes, and multicellular organismal processes. There were 10 IbLBD proteins that can participate in these three processes. Under the cellular component category, the most highly enriched categories were related to cell, cell part, and organelle. Among them, 11 genes were annotated separately, indicating that these 11 genes play an important role in the composition of these three categories. Regarding molecular function, there were only four IbLBD proteins having the capacity to bind to other molecules; additionally, it also was the only GO term in molecular function. The top 20 GO terms are shown in Figure 13. The strongest enrichment and the highest enrichment factor (800.33) were observed for the process of adventitious root development, followed by the process of lateral root formation (442.29). In addition, the largest number of genes (nine) was associated with the GO term “anatomical structure development”.

## 4. Discussion

The gene families of sweet potatoes have been extensively explored using genome sequencing. In this study, in the sweet potato genome, we found 53 *IbLBD* genes (Table 1), which were divided into two major classes and six subclasses (Ia~Id, IIa, and IIb). There was a considerable variance in LBD protein family members across species, and diploid plants often contained fewer *LBD* genes, for instance, *Arabidopsis*, tomato [54], rice [8], and corn [9], whereas other diploid plants contained fewer than 50 *LBD* genes. Contrarily, the number of *LBD* genes in polyploid plants was much larger. For instance, 90 *LBD* genes were found in soybeans [13] and wheat [14]. Although the sweet potato is an allohexaploid plant, the number of identified *LBD* gene families was less than in most polyploid plants. This may be because the differences between the two genomes that comprise the sweet potato were insignificant [55]. Meanwhile, there are some heterogeneities between sweet potato chromosomes, so it is difficult to assemble the genome, thereby inducing incomplete sequencing results. Results from the gene structure analysis revealed that most *LBD* genes had less than four exons, which was similar to the findings related to angiosperms such as *Arabidopsis* and rice. After the phylogenetic tree was constructed, all *IbLBD* genes were categorized into Class I or Class II, comprising 45 and 8 *LBD* genes, respectively. We found that the number of *LBD* genes in Class I of *Arabidopsis*, moso bamboo [56], and pears [57] was high in Class II, which was consistent with previous studies. There was also a difference in the Class I classification. Specifically, Class I in the common bean was subdivided into 11 subclasses [58], while wheat contained eight subclasses. However, there were relatively similar classification results for sweet potato, passion fruit, and moso bamboo.

Performing a collinearity analysis within a certain species reveals the homology of genes among different chromosomes. The chromosome localization and collinearity analysis results confirmed that *IbLBD* contains 17 duplicated gene pairs, including 3 tandem duplicated gene pairs and 14 segmental duplicated gene pairs. Therefore, we speculated that in the evolution process, the *LBD* gene family expansion was dominated by the segmental duplication mechanism and supplemented by the tandem duplication mechanism. This contributed to the development of novel gene functions and may also explain the comparatively conservative number of *IbLBD* gene family members. A collinearity analysis among different species also exposes gene evolution and genetic relationships among them. The evolutionary relationship between sweet potato and other species was explored from the perspective of *LBD* family genes through a collinear analysis. The results indicated that the *LBD* family genes of sweet potato were more closely related to the same Solanales plant, tomato and pepper. Furthermore, a total of 52 and 50 collinear gene pairs were found in these two plants, respectively. However, the genetic relationship between sweet potato and gramineous plants (maize and rice) was insignificant with only a few collinear gene pairs. These findings were consistent with the results of the genetic relationship analysis.

The cis-acting element of promoters is critical in gene expression control. Okushima et al. reported that *LBD16/ASL18* in *Arabidopsis* participated in lateral root formation and growth hormone response [16]. Louise et al. found that *AtLBD20* contributed to the plant disease resistance process mediated by COI-dependent jasmonate (JA) [26]. In this study, we verified that the *LBD* promoter region of sweet potato contained several elements related to the hormone regulation pathway. Moreover, the photoresponsive elements, abscisic acid-responsive elements, methyl jasmonate-responsive elements, and drought-inducing elements at the MYB binding sites were the most extensively distributed. Among them, the photoresponsive elements and abscisic acid-responsive elements were detected in most genes. Therefore, we inferred that light and abscisic acid may influence *IbLBD* gene expression, thereby affecting the growth and development of sweet potato. More specifically, *IbLBD37*, *IbLBD9*, and *IbLBD12* had seven, six, and six ABRE elements, respectively. These elements may be involved in the regulation of ABA metabolism in sweet potato. Huang et al. [56] discovered that all *LBD* gene promoters in moso bamboo contained drought-inducing elements at the MYB binding sites. These drought-inducing elements were detected in 22 sweet potato genes with certain differences in quantity. Results from the heatmap analysis revealed that *IbLBD6*, *IbLBD24*, and *IbLBD48* were highly expressed under drought stress, while *IbLBD41* in Class I and *IbLBD2* in Class II were highly expressed under both abiotic stresses. This may be related to the internal cis-acting elements that were involved in defense and stress responses. These findings indicated that *LBD* gene expression in sweet potato was regulated by cis-elements related to plant development and abiotic stress tolerance.

We discovered that 25% of the *IbLBD* genes were highly expressed in sweet potato stems when we analyzed the expression patterns of the *IbLBD* genes, which was consistent with the findings related to turnips [11], cotton [29], and pears [57]. Additionally, *StLBD3-5* and *StLBD2-6* were highly expressed in potato stems under drought stress [28], which improved their drought resistance. This was consistent with the expression of *IbLBD25* under drought stress. Furthermore, *Bra035860* was relatively highly expressed in turnip roots and stems [11]. The results of the evolutionary analysis suggested that *IbLBD47*, which had high homology with *Bra035860*, was specifically expressed in the stem. As a result, we speculated that the *IbLBD* genes were principally involved in the functions of plant stem development and they were also highly expressed in flowers and fruits. As reported in a previous study, *AtASL1* regulated flower development in *Arabidopsis* by controlling the development orientation of petal cells [59]. The expression of *Solyc01g091420.1.1*, *Solyc03g119530.1.1*, and *Solyc04g050010.1.1* was also upregulated in tomatoes between the flowering and ripening periods [54]. According to the findings of this research, *IbLBD17* and *IbLBD28* were also shown to be highly expressed in sweet potato flowers and fruits but not in other parts of the plant. Therefore, we concluded that these genes had a role in developing sweet potato flowers and fruits. Moreover, LOB was found to be expressed in the base of lateral roots [60]. The findings of the heatmap analysis indicated that all of the highly expressed genes in the primary roots belonged to Class I. This further demonstrated that the *LBD* genes in Class I were mainly involved in the development of the lateral roots in plants, consistent with the results of GO enrichment analysis studies. More than half of the top 20 enriched GO terms in the *IbLBDs* were related to plant organ development and formation, among which morphogenic functions involving lateral root formation and root development were the most significant. However, *IbLBD* genes were expressed at a lower level in the foliage, which was consistent with the findings of research on the vast majority of plants, including cotton, common beans, and pears.

Under environmental stress, *IbLBD* genes also assert a regulatory role. Usually, the root system is the first part to be affected by environmental stress. As the heatmap shows, *LBD* genes were commonly significantly expressed in the primary root under diverse stress. Most of the promoters of *IbLBD* genes contained MBS elements, indicating that these genes played a regulatory role in plants under drought stress. Among them, 11 *IbLBD* genes were upregulated under salt and drought stresses. These genes are likely to respond to salt and drought stress, playing a crucial role in regulating stress responses. They may also exhibit functional redundancy. The tissue expression analysis results confirmed that different *LBD* family members had specific tissue expression patterns in the aspects of temporal and spatial expression. Furthermore, some *LBD* family members exhibited functional specificity. The expression of some *LBD* genes varied for different root morphological profiles, therefore affecting sweet potato root systems differently. However, it is necessary to further clarify the specific function of these genes. Generally, genes with high homology have similar functions. Under salt stress, the expression of *IbLBD46* and *IbLBD47* was downregulated in different parts of sweet potato. This was highly homologous with *Solyc02g087570.1.1* and *Solyc03g112430.1.1* in tomatoes [54], which exhibited similar results. Under drought stress, the expression of *StLBD26* was upregulated in potato stems [28], which was consistent with the results of *IbLBD25*. As the core signal transduction component in *Arabidopsis*, *LBD15* regulated the water loss rate of plants through stomata. Meanwhile, the overexpression of *LBD15* enhanced the drought resistance of plants [61]. Additionally, there was high homology between *AtLBD15* and *IbLBD21*, while *IbLBD21* was highly expressed under drought stress. The PCR findings demonstrated that *IbLBD21* expression under drought stress was identical to the transcriptome data, suggesting a similar function.

## 5. Conclusions

In this study, we systematically analyzed and identified 53 *IbLBD* genes in sweet potato, which were dispersed across 13 chromosomes and classified into 6 categories. According to studies of sweet potato evolution, segment duplication may have a more significant impact than tandem duplication in expending *LBD* gene growth in the sweet potato genome. Collinear analysis verified the significant degree of similarity between the sweet potato and plants in the Solanales family. Based on the RNA-seq data, the tissue and abiotic stress of *IbLBD* gene expression pattern was revealed, which was also supported by quantitative analysis and GO annotation. It was discovered that most *LBD* genes are expressed far more highly in some tissues than others: some gene expression levels were very high in specific tissues such as *IbLBD30/39/43*, which were specifically expressed in the fruit, while *IbLBD47* was only highly expressed in the stem, indicating a special function in the development of some tissues. The qRT-PCR was conducted to certify the expression levels of twelve *IbLBDs* in diverse tissues and different abiotic stresses. Under drought stress, *IbLBD21* and *IbLBD15* were upregulated in the stem and had a high degree of homology between them. The PCR results demonstrated that the expression of *IbLBD21* under drought stress was consistent with the transcriptome data. Further research will be conducted on *IbLBD21*. Overall, we believed that *IbLBD2*, *IbLBD7*, *IbLBD12*, and *IbLBD21* genes in sweet potato have important research value. These results help explain *IbLBD* gene family formation. Research to study the biological process of *IbLBD* transcription factors under different stress conditions is required to further investigate the functions of sweet potato *LBD* genes.

## Figures and Tables

**Figure 1 genes-15-00237-f001:**
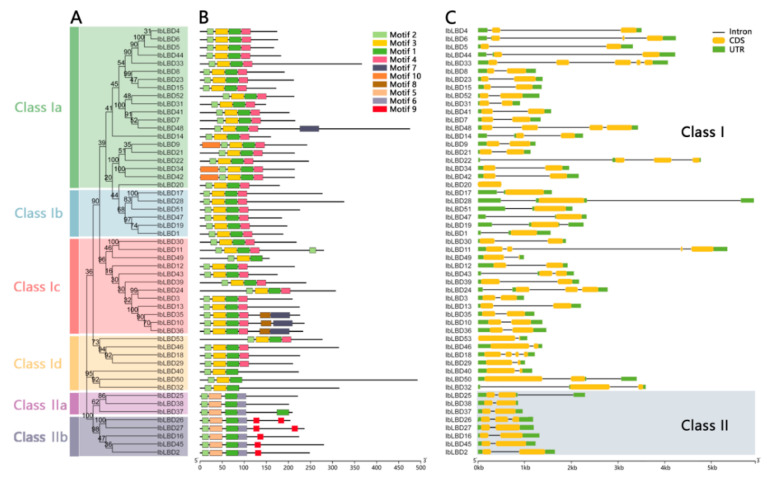
Phylogenetic tree, conservative motif, and gene structure of *IbLBD* family in sweet potato. (**A**) A maximum likelihood (ML) phylogenetic tree of sweet potato protein with 1000 bootstrap replicates was constructed based on the full-length sequence in MEGA11. (**B**) Distribution of conservative motifs in IbLBD proteins with colored boxes representing motifs 1–10 and scale representing 50 amino acids. (**C**) The genetic structure of the *IbLBD* gene, including intron (black line), exon (yellow rectangle), and untranslated region (UTR, green rectangle), with scale representing 1 kb.

**Figure 2 genes-15-00237-f002:**
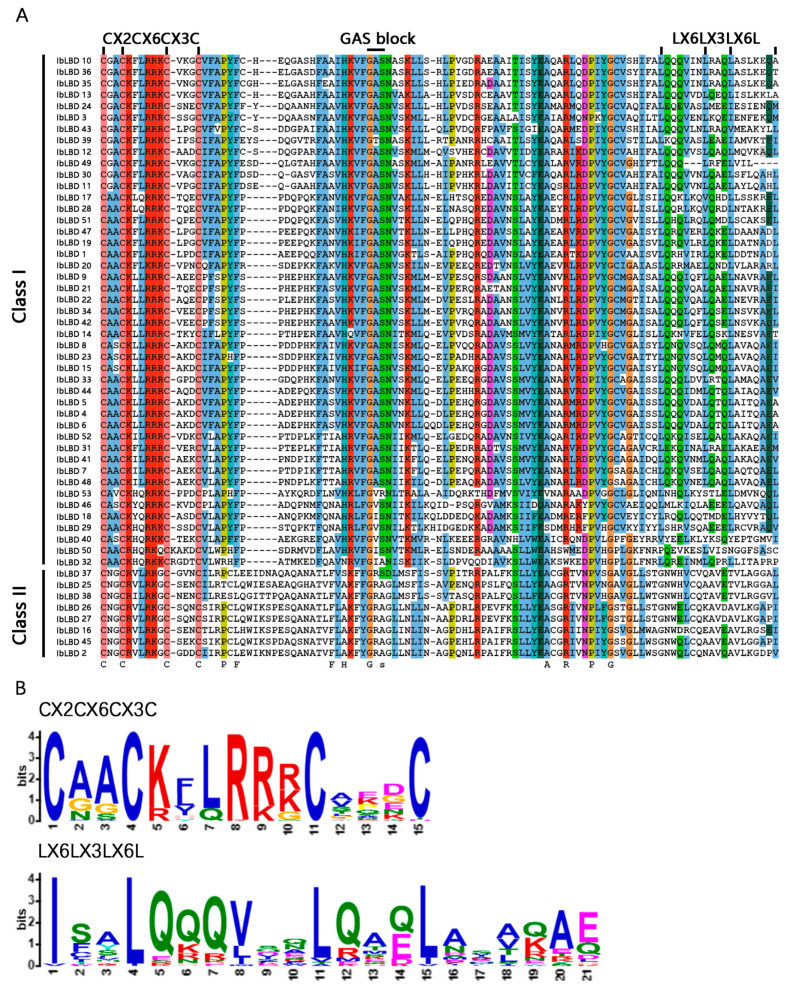
Multiple sequence alignment and conservative domains of sweet potato IbLBD proteins. (**A**) The zinc finger domain (CX2CX6CX3C) was present in all 53 predicted IbLBD protein sequences, while the leucine zipper-like motif (LX6LX3LX6L) was present only in Class Ⅰ IbLBD proteins. (**B**) Alignment of conservative motifs generated by MEME online website for the two protein domains.

**Figure 3 genes-15-00237-f003:**
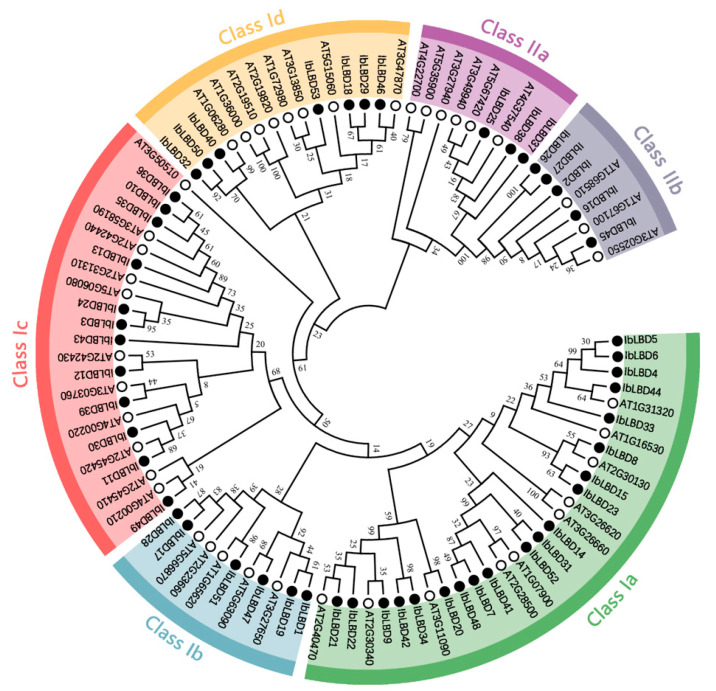
Phylogenetic trees of LBD proteins for sweet potato and *Arabidopsis*. *Arabidopsis* LBD protein sequences were downloaded from the TAIR database. A phylogenetic tree was constructed by the maximum likelihood method based on MEGA11 with 1000 bootstrap replicates performed. The tree was divided into six subfamilies represented by outer rings with different colors; black circles and white circles represent the sweet potato and *Arabidopsis LBD* genes.

**Figure 4 genes-15-00237-f004:**
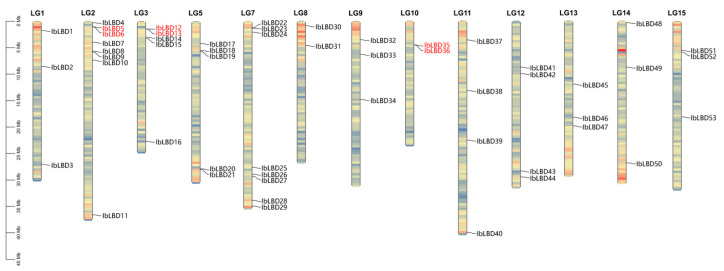
Locations of sweet potato *LBD* genes on chromosomes. The basic unit indicated a chromosome length of 5.0 Mb. For each chromosome, the number was labeled on the upper side with red indicating a gene pair with tandem duplication.

**Figure 5 genes-15-00237-f005:**
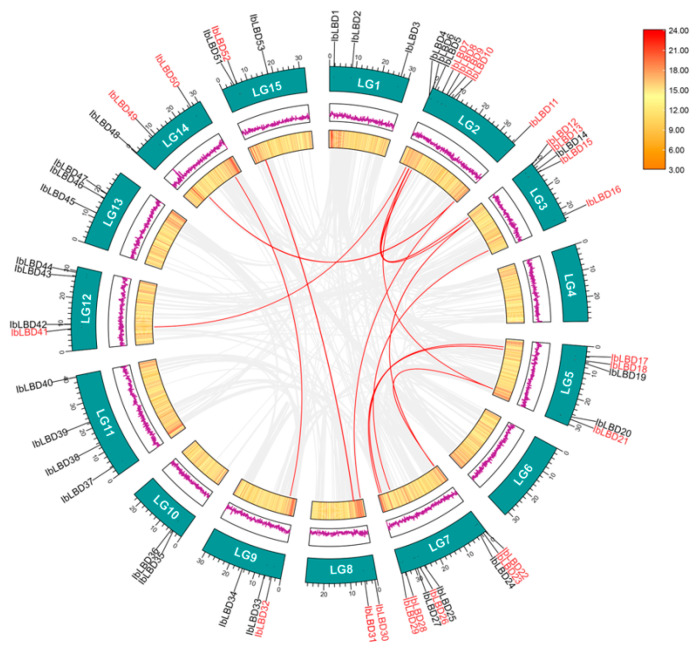
Distribution and collinearity of the *IbLBD* gene family in the sweet potato genome. *IbLBDs* labeled with red had collinearity, while those labeled with black had no collinearity. The two rings in the middle represented the gene density of each chromosome. The gray background lines represent collinear background and the red lines indicate a collinearity relationship between *IbLBD* members.

**Figure 6 genes-15-00237-f006:**
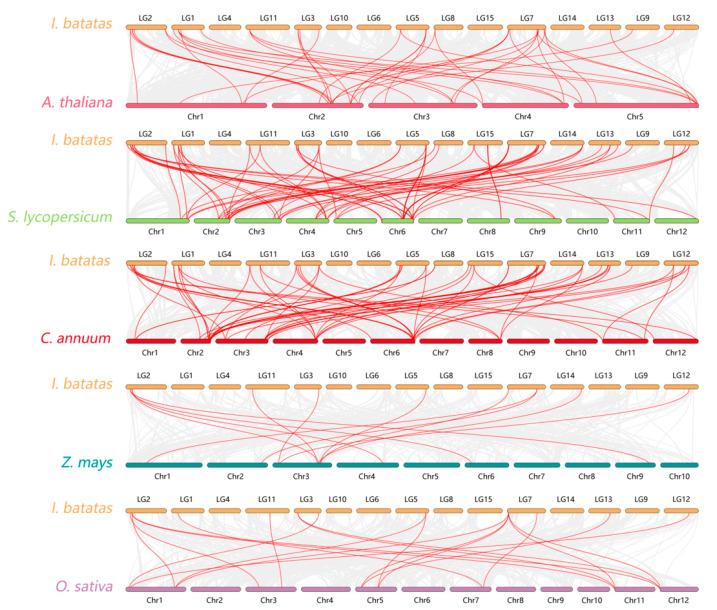
Collinearity analysis of LBD protein in sweet potato among species. The species were *Arabidopsis*, tomato, capsicum, maize, and rice. The red line represents the homologous *LBD* gene pair of the plant genome and the gray line represents the collinear block of the plant genome.

**Figure 7 genes-15-00237-f007:**
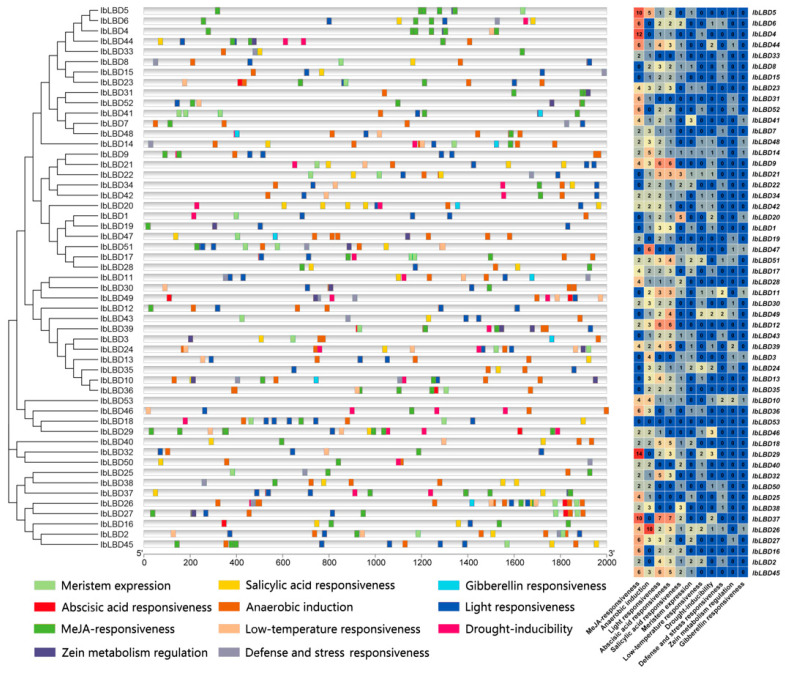
Distribution of cis-acting elements of *IbLBD* gene family in sweet potato. Distribution of cis-acting elements identified in the 2000 bp upstream promoter region of sweet potato *IbLBD* gene and the number of cis-acting element types for the *IbLBD* gene on the promoter.

**Figure 8 genes-15-00237-f008:**
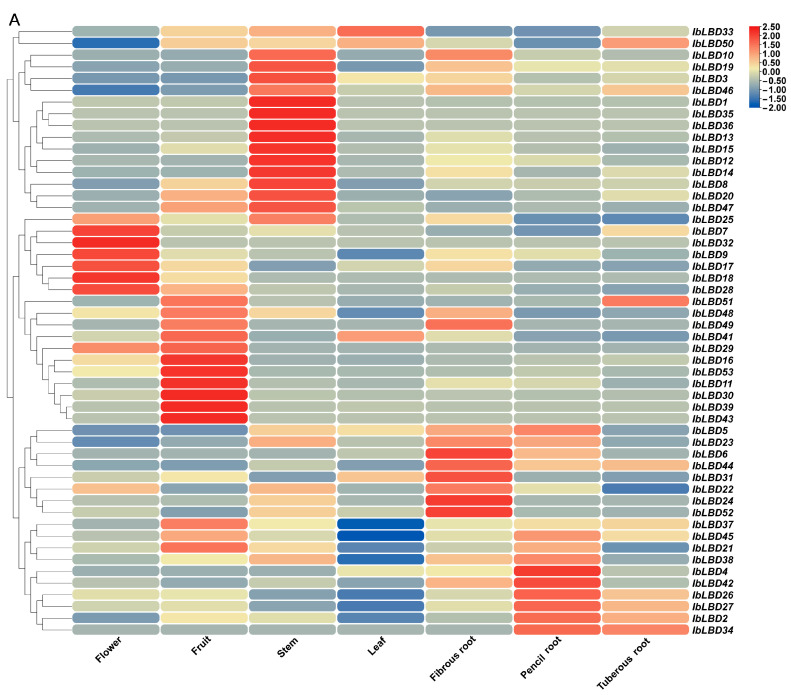
Expression patterns of the *IbLBD* gene family in sweet potato. (**A**) *IbLBD* gene expression heatmap in various sweet potato tissues. (**B**) Expression heatmap of sweet potato root, stem, and leaf tissues induced by salt and drought. Red and blue indicated the intensity of genes in the heatmap: the more intense the red, the higher the gene expression level, and the more intense the blue, the lower the gene expression level.

**Figure 9 genes-15-00237-f009:**
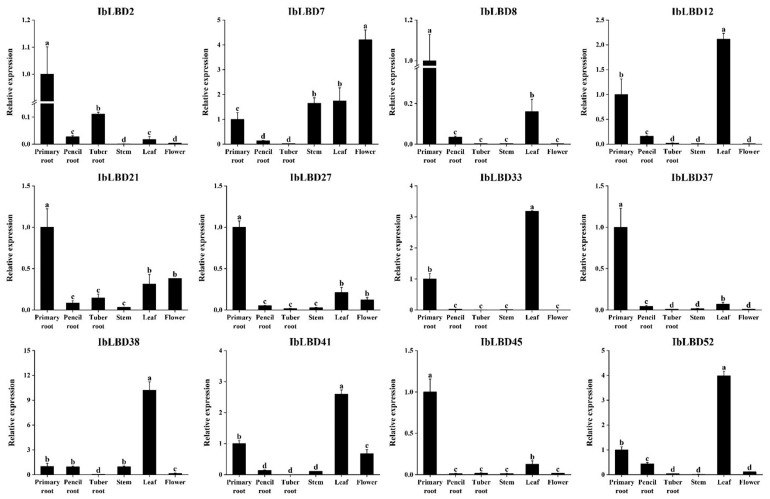
Expression patterns of twelve *IbLBD* genes in different tissues. The x axes represent different tissues including primary root, pencil root, tuber root, stem, leaf, and flower; the y axes indicate the relative expression of *IbLBD* genes. The different letters of a, b, c, and d indicate significant differences at *p* < 0.05, as determined by one-way ANOVA with SPSS single factor tests.

**Figure 10 genes-15-00237-f010:**
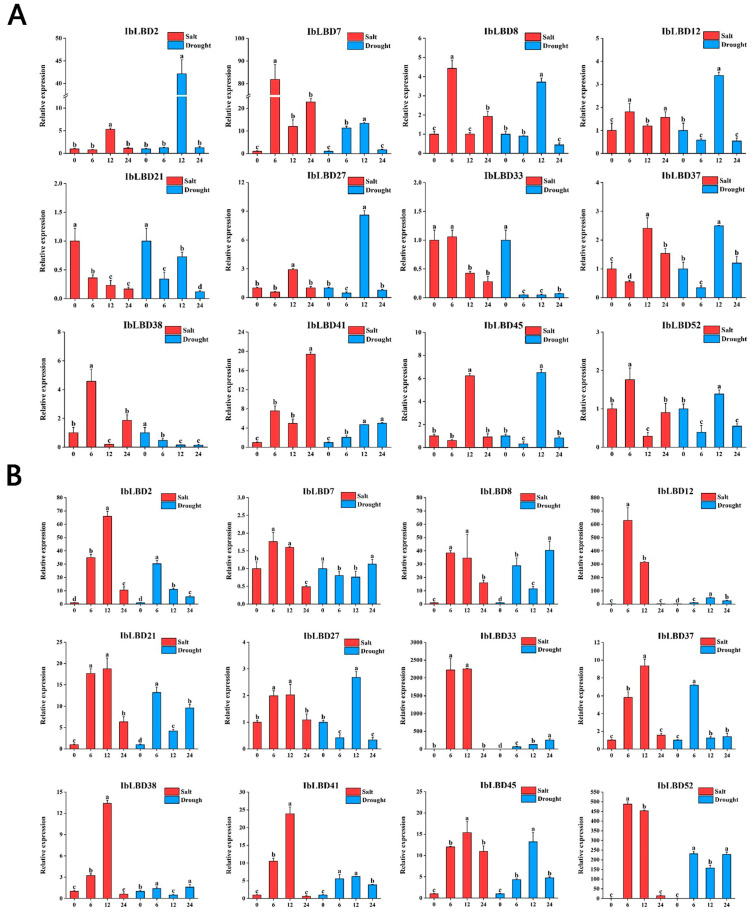
Changes in the expression levels of twelve *IbLBD* genes in different tissues under salt and drought treatments. (**A**) 12 *IbLBD* gene expression levels in primary root under salt and drought treatments. (**B**) 12 *IbLBD* gene expression levels in stem under salt and drought treatments. (**C**) 12 *IbLBD* gene expression levels in leaf under salt and drought treatments. The different letters of a, b, c, and d indicate significant differences at *p* < 0.05, as determined by one-way ANOVA with SPSS single factor tests.

**Figure 11 genes-15-00237-f011:**
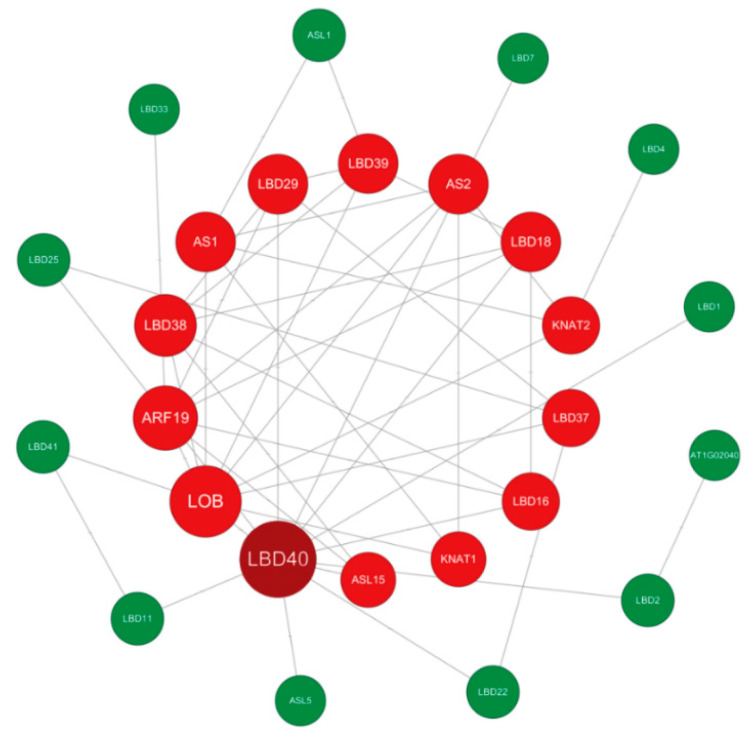
Sweet potato IbLBD functional interaction networks based on *Arabidopsis* orthologs. Proteins serve as network nodes and protein–protein relationships are represented by lines. The size of the node denotes the number of proteins that interacted with one another.

**Figure 12 genes-15-00237-f012:**
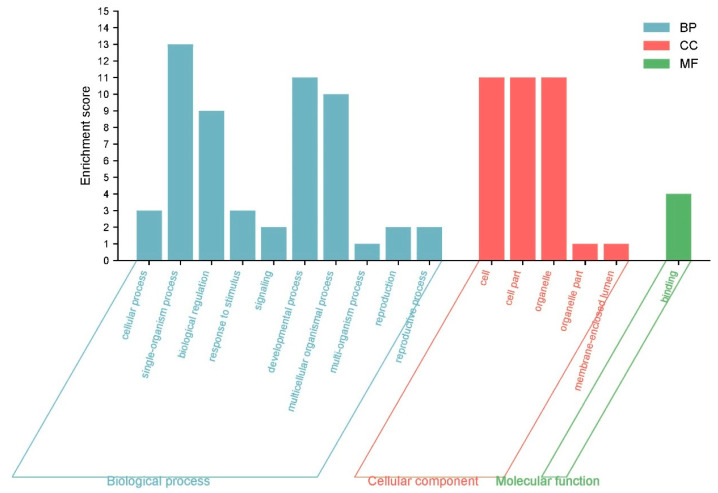
GO annotations for IbLBD proteins. The GO annotation is divided into three main categories: biological process, cellular component, and molecular function. The y axis represents the enrichment score of the number of genes in each category.

**Figure 13 genes-15-00237-f013:**
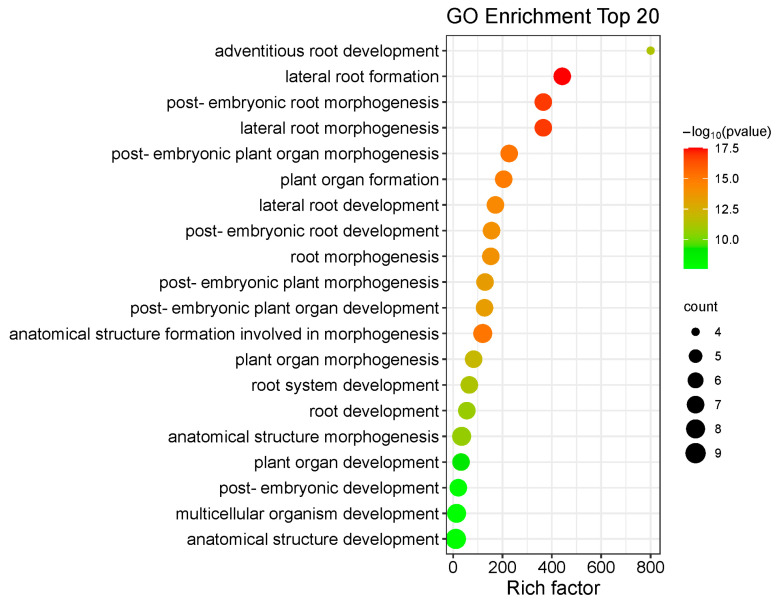
The top 20 enriched GO terms of candidate *IbLBD* target genes. The black circles indicate the number of target genes, and different colors indicate the −log_10_ (*p*-value), ranging from 0 to 17.5.

**Table 1 genes-15-00237-t001:** Identification of *LBD* genes and analysis of physicochemical properties of proteins in sweet potato.

Gene Name	Accession Number	Chromosome Location		Size (aa)	Mw (kD)	pI	GRAVY	Predicted Location
*IbLBD1*	OR133645	1687306	1688873	188	20.72	6.5	−0.311	Nucleus
*IbLBD2*	OR133646	8494464	8496120	249	27.22	6.6	−0.447	Nucleus
*IbLBD3*	OR133647	27014760	27015755	209	23.87	5.21	−0.431	Nucleus
*IbLBD4*	OR133648	271534	275058	174	19.07	8.97	−0.411	Nucleus
*IbLBD5*	OR133649	1039868	1043206	167	18.27	7.72	−0.411	Nucleus
*IbLBD6*	OR133650	1069879	1074143	191	21.44	10.08	−0.717	Nucleus
*IbLBD7*	OR133651	3938571	3939920	215	23.25	6.14	−0.039	Nucleus
*IbLBD8*	OR133652	5639554	5640802	191	20.68	6.99	−0.117	Nucleus
*IbLBD9*	OR133653	5829105	5830345	242	25.65	7.66	0.173	Nucleus
*IbLBD10*	OR133654	7327214	7328603	236	26.03	6.2	−0.534	Nucleus
*IbLBD11*	OR133655	36568489	36573862	334	35.22	9.28	−0.256	Nucleus
*IbLBD12*	OR133656	1456672	1458605	214	23.11	8.36	−0.817	Nucleus
*IbLBD13*	OR133657	1466969	1469187	225	24.32	6.48	−0.534	Nucleus
*IbLBD14*	OR133658	3005283	3007545	160	17.77	6.71	0.322	Nucleus
*IbLBD15*	OR133659	3278659	3280029	172	18.94	7.63	−0.211	Nucleus
*IbLBD16*	OR133660	22684760	22686086	224	24.46	8.6	−0.628	Nucleus
*IbLBD17*	OR133661	4128151	4129743	277	31.01	6.16	0.000	Nucleus
*IbLBD18*	OR133662	5451750	5452974	226	25.48	4.68	−0.700	Nucleus
*IbLBD19*	OR133663	5662166	5664439	197	22.23	8.15	−0.278	Nucleus
*IbLBD20*	OR133664	27801512	27802278	180	19.99	8.28	−0.445	Nucleus
*IbLBD21*	OR133665	27967462	27968594	214	23.51	8.6	−0.445	Nucleus
*IbLBD22*	OR133666	1121586	1126383	246	26.19	8.67	0.122	Nucleus
*IbLBD23*	OR133667	1261129	1262521	212	23.44	5.79	−0.584	Nucleus
*IbLBD24*	OR133668	2016696	2019488	307	34.00	4.58	−0.080	Nucleus
*IbLBD25*	OR133669	27558292	27560598	187	20.18	9.27	−0.284	Nucleus
*IbLBD26*	OR133670	28842217	28843398	204	21.82	8.17	−0.417	Nucleus
*IbLBD27*	OR133671	29266026	29267217	236	25.74	8.52	−0.528	Nucleus
*IbLBD28*	OR133672	33791108	33797053	326	36.47	6.66	−0.462	Nucleus
*IbLBD29*	OR133673	34877261	34878276	210	23.58	6.65	−0.745	Nucleus
*IbLBD30*	OR133674	850688	852583	218	22.77	8.47	−0.250	Nucleus
*IbLBD31*	OR133675	4610655	4611562	149	16.16	8.75	−0.017	Nucleus
*IbLBD32*	OR133676	3472142	3475757	322	36.23	9.58	−0.739	Nucleus
*IbLBD33*	OR133677	6217307	6221397	366	39.29	4.59	−0.306	Nucleus
*IbLBD34*	OR133678	14820911	14822874	214	23.46	9.11	−0.261	Nucleus
*IbLBD35*	OR133679	4501947	4503162	226	24.89	5.39	−0.273	Nucleus
*IbLBD36*	OR133680	4532985	4534460	245	27.27	4.97	−0.233	Nucleus
*IbLBD37*	OR133681	3579351	3580312	209	22.47	8.67	−0.101	Nucleus
*IbLBD38*	OR133682	13130843	13131707	201	22.17	7.64	−0.336	Nucleus
*IbLBD39*	OR133683	22497885	22500062	240	26.20	5.47	−0.062	Nucleus
*IbLBD40*	OR133684	39950925	39952093	223	25.26	8.77	−0.966	Nucleus
*IbLBD41*	OR133685	8697591	8699166	202	21.89	6.11	−0.084	Nucleus
*IbLBD42*	OR133686	9909480	9911649	214	23.43	9.25	−0.284	Nucleus
*IbLBD43*	OR133687	28316625	28318692	175	18.91	6.71	−0.007	Nucleus
*IbLBD44*	OR133688	29434436	29438687	183	19.95	8.58	−0.578	Nucleus
*IbLBD45*	OR133689	11950627	11951867	280	30.17	5.64	−1.033	Nucleus
*IbLBD46*	OR133690	18216174	18217557	314	34.94	5.46	−0.580	Nucleus
*IbLBD47*	OR133691	19821743	19824082	185	20.18	6.88	−0.728	Nucleus
*IbLBD48*	OR133692	291788	295234	475	52.48	6.25	0.084	Nucleus
*IbLBD49*	OR133693	8713728	8714718	157	17.29	9.97	0.361	Nucleus
*IbLBD50*	OR133694	26782324	26785743	520	57.51	8.81	−0.730	Nucleus
*IbLBD51*	OR133695	5492804	5494838	227	24.66	8.56	−0.495	Nucleus
*IbLBD52*	OR133696	5850533	5851542	213	22.89	5.47	−0.056	Nucleus
*IbLBD53*	OR133697	18062560	18063739	277	31.14	9.69	−0.403	Nucleus

## Data Availability

Data are contained within the article or Appendix A.

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
