# Peer review of "Genome-Wide Analysis of the Lateral Organ Boundaries Domain (LBD) Gene Family in Sweet Potato (Ipomoea batatas)"

_genes, 2024, doi:10.3390/genes15020237_

Round 1
Reviewer 1 Report
Comments and Suggestions for Authors
1. Please be direct in stating the objective of this study at final part of the Introduction. In the present form it appears to be a summary of materials and methods.
2. Materials and methods. Lines 70-71. The website provided for the Ipomoea Genome Hub (https://ipomoea-genome.org/) cannot be accessed. Please provide an active URL.
3. The page where the Markov model was obtained for the LBD family pops up with the message: “The pfam.sanger.ac.uk website is no longer supported at The Wellcome Sanger Institute”. Please provide an alternative website.
4. The webpage http://wwwcsbio.sjtu.edu.cn/bioinf/Cell-PLoc-2/ to predict subcellular localization of proteins is no longer active.
5. The online software (http://gsds.gao-lab.org/) used for predicting exon-intron structure is no longer available.
6. Lines 171, 271. Common names are not written in italics.
7. Conclusions. Lines 560-561. Authors found 53 lbLBD genes, rather than 53 gene families.
Author Response
Dear Reviewer,
Firstly, all authors would like to thank you for reviewing the content of our manuscript. We have made corresponding modifications to the questions you raised, and you can find the modifications we made in document "genes-2845666 point-by-point response(Reviewer 1 content modification)". It should be noted that the content marked in yellow is the part of the original text that needs to be modified, while the parts highlighted in green are the modifications we made based on the questions you raised. In certain aspects of content, we have also made corresponding adjustments to the references. During your review process, If we have any modifications that are not in place or if you have any questions, we hope you can contact us and we will respond positively. Once again, we appreciate your review of the manuscript.
Wishing you have a happy work and life experience!
Sincerely,
Lei Shi

Reviewer 2 Report
Comments and Suggestions for Authors
The authors decribed LBD gene structure in sweet potato.
Authors made a good work, however, significant corrections are require.
Please, split long sentece, remove repetions etc. Some points are below:
Line 23: what do you mean as various root?
Lines 27: proteins are proteins?
Line 29: „stem of Arabidopsis (Arabidopsis thaliana) seedlings“ ?? Seedlings does have a stem. Stem come in the adult plants.
Line 64: „biological information methods“ = bioinformatics methods
Line 66: „To provide a theoretical foundation for future study on the functional characterisation of the sweet potato LBD gene family.“ ?? Not full sentence??
Lines 41- 46: plesae, split sentence to several.
Line 48: „inhibiting tolerance under drought stress“ ?? Maybe reduce tolerance?
Line 55: „protein bioinformatics“ ??
Line 59: „resource of industrial“ ??
Line 128- frozen, line 136 . cooled. What the differences?
I am not sure it is a good idea to use nutrient starvation fir 10 days before stress. Please, explain why?
Lines 143- 144_ looks like unclear. Please, clarify.
Line 166: „greater“?? = Higher. Greater have different meaning.
Line 271: „To further investigate the evolutionary relationships between the LBD family and other species“ ¿?? Sweet potato word missing.
Line 294: „zeatin metabolic regulatory elements“ ? Why in prevous senteobe to was GA, ABA responsive elemnet, but here you choose zeatin? Not zeatin-ribosid, not oither cytokiinin?
Line 333: „A total of 45 IbLBD genes were“ ¿?? Maybe you mean up-regulation of gene expression? Or you mean that 8 genes were eliminated from genome?
Line 339_ greater = higher!
Line 345: „genes had a protective mechanism for roots“ ¿?? This is completely wrong. Gene itself can not have a protective mechanism. Moreover, for ptrove that higher gene expression play a role in protection you need to have a lines without functional gene and prove that gene functionality is a key for protection. You can only conclude that gene up-regulated under stress.
Line 357 „Among the genes, the expression analysis genes“ ¿??
Lines 412- 414: one sentence was repeated twice.
Comments on the Quality of English Language
Many long sentences with multi messages, some repetitions etc.
Author Response
Dear Reviewer,
Firstly, all authors would like to thank you for reviewing the content of our manuscript. We have made corresponding modifications to the questions you raised, and you can find the modifications we made in document "genes-2845666 point-by-point response(Reviewer 2 content modification)". It should be noted that The content marked in yellow is the unmodified part of the original text, while the parts highlighted in blue are the modifications we made based on the questions you raised. In certain aspects of the content, we have made modifications to the references and areas with similar issues. During your review process, If we have any modifications that are not accurate enough or if you have any questions, we hope you can contact us and we will respond positively. Once again, we appreciate your review of the manuscript.
Wishing you have a happy work and life experience!
Sincerely,
Lei Shi

Round 2
Reviewer 2 Report
Comments and Suggestions for Authors
Thank you! All correction almost OK, except several:
"improved drought sensitivity" ?? Sensitivity can not improved. It can only increase!
Lines 171- 182: This part is very mixing. Please, clarify conditions of plant growth at the first part, and all process of freezing, etc in the second.
You did not explain what the reason to keep plants 10 days in clean water (starvation from nutrients) before stress.
The rest is OK.
Author Response
Dear reviewer,
All authors, thank you again for your further review and suggestions for revision of our manuscript. We have rewritten the content of the experimental process and the highlighted blue part is the modified content. If you have any questions about the current conttent or format, we will make further revisions and actively respond to your suggestions.
Wishing you have a happy work and life experience!
Sincerely,
Lei Shi
